# Twelve Weeks of Web-Based Low to Moderate Physical Activity Breaks with Coordinative Exercises at the Workplace Increase Motor Skills but Not Motor Abilities in Office Workers—A Randomised Controlled Pilot Study

**DOI:** 10.3390/ijerph20032193

**Published:** 2023-01-25

**Authors:** Carina Scharf, Markus Tilp

**Affiliations:** Institute of Human Movement Science, Sport and Health, University of Graz, 8010 Graz, Austria

**Keywords:** physical activity break, juggling, balance training, motor fitness, work health promotion, adult

## Abstract

Integrating physical activity interventions at the workplace can have positive effects on the employees’ health. This study aimed to evaluate a physical activity break with coordinative exercises (PAB) including juggling and balance tasks and to assess its effects on motor abilities. Thirty-two university employees were randomly allocated to an intervention (IG:20) or a control (CG:12) group. The IG participated two times per week for 12 weeks in a PAB with a duration of 15 to 20 min. We measured the unimanual, bimanual finger, and hand dexterity with the Purdue Pegboard Test, the reaction time with the Fall Stick Test, and the dynamic balance with the Y Balance Test. Juggling performance was assessed by measuring the time(s) of performing a three-ball-cascade. Furthermore, an evaluation of the PAB was executed. Participants in the IG improved their juggling performance after six and twelve weeks. These increases were significantly different compared to the CG. However, no other parameters changed significantly. The evaluation showed that the PAB was enjoyable and led to subjective improvements in the participants health and working routine. To conclude, PAB can lead to improvements in juggling performance, subjective health, and the working routine.

## 1. Introduction

Regular physical activity can have a positive impact on the health of adults. Although this is already commonly known, one in four adults does not meet the global recommendations for physical activity for health. Furthermore, there is a trend of an increasing inactivity in economically developed countries. Inactivity causes 1 to 3% of national health care costs, and higher costs can be assumed because this calculation does not include costs associated with musculoskeletal conditions and mental health [1].

To increase physical activity [2,3] and reduce sedentary behaviour in adults, the World Health Organisation (WHO) [1] recommends integrating physical activity at the workplace. The workplace can be a key setting to reach this age group, since the largest group of adults are employed [4] and are spending about half of their waking hours during weekdays at the workplace [5]. Besides the positive physical and mental health effects [5,6,7,8,9,10] that come along with being physical active at the workplace, physical activity interventions can have a positive impact on absenteeism [11]. Moreover, physically active employees are more productive than less active employees [12].

A physical activity break is an opportunity to include physical activity at the workplace [1]. Bramante et al. [13] confirmed that physical activity breaks are feasible at the workplace and desired by the employees. The employees answered a questionnaire after participating in a 10-min physical activity break. They indicated that practicing physical activity breaks regularly would increase comfort and productivity at work. Furthermore, they were very positive about the feasibility and enjoyment of the physical activity breaks. In some studies, the effects of different physical activity breaks on different outcomes have already been investigated. For example, aerobic exercises like walking [14,15,16], running, cycling or rowing [17] increased physical activity [14] and improved mental health [14,16,17], and cardiovascular health [15]. A physical activity break—which included resistance exercises for four months, three times per week, for 20 min—resulted in increases in physical activity, physical fitness, muscle strength and productivity, and in a reduction of perceived fatigue in dairy industry workers [18]. Improvements in strength and postural balance were found in a study by Jørgensen et al. [19] after 12 weeks of resistance and balance exercises, which lasted 20 min and were offered three times per week. Taylor et al. [20] implemented ”Booster Breaks“, which consisted of aerobic, resistance, stretching and relaxing exercises for 15 min. After six months, the intervention group increased daily step counts, decreased sedentary behaviour, and maintained their BMI-status, but also decreased their leisure time activities. Another study, where a combination of different kinds of exercises were included in a physical activity break, was conducted by Michishita et al. [21,22]. They implemented 10-min lunch fitness breaks, which consisted of aerobic, resistance, and stretching exercises, and cognitive functional training. After 10 weeks of training, the participating workers improved their physical activity, mental health, job satisfaction, interpersonal stress and presenteeism [21,22]. The acute effects of a 10-min session of a combination of mobilisation, stabilisation, and resistance training, or of motor-cognitive coordination training, were investigated in a study by Niederer et al. [23]. They found increases in both groups regarding attention and concentration. A 12-week video- and web-based physical activity break with Qigong exercises, practiced by sedentary office workers, increased physical activity and comfort, reduced the increase of sedentary hours during the week, and did not influence work performance negatively [24].

Despite these interesting findings, there are still some research gaps, e.g., if and how integrating light-intensity physical activity and breaking up sedentary time at the workplace can influence the health of adults [25]. Coordinative training offers the opportunity to integrate low to moderate physical activity into the office day, which can also be practiced by unactive adults without sweating, a factor that often impedes participation in physical activity breaks at the workplace. Coordinative training can also increase motor competence, a lack of which has been shown to be related to compromised health-related fitness [26]. However, the effects of a physical activity break with coordinative exercises at the workplace on motor abilities has not been investigated yet. Furthermore, we have learned from the COVID-19 pandemic that web-based exercise training videos appear as new, attractive and necessary opportunities to increase physical activity and fitness [27]. To our knowledge, so far only one study used a web- and video-based physical activity break at the workplace [24]. 

For this reason, the aim of the present pilot study was to evaluate a three-month web-and video-based physical activity break with coordinative exercises and assess its effects on motor abilities in young and middle-aged adults. To investigate these effects, a physical activity break was implemented as a workplace health promotion project for university employees.

## 2. Materials and Methods

### 2.1. Participants

A total of 55 participants were recruited via email and via advertisement on the employees’ website of the University of Graz (Austria). The exclusion criteria for the recruitment were aged younger than 20 years and older than 65 years, regular participation in intense coordinative and/or motor training (e.g., juggling, playing the piano), and cardiovascular, psychiatric, and/or neurological diseases. After receiving further detailed information on the study procedure, 41 participants took part in the familiarization session. Due to time management, five participants dropped out before the first measurement. Subsequently, 36 participants started with the study. The participants were matched in pairs (gender and age) and were randomly allocated, at the rate of 2:1, to the intervention (IG) or the control group (CG). The higher allocation to the intervention group was used to include enough participants who would perform the physical activity break. During the intervention period, a further three participants dropped out of the study. Only participants who completed at least 2/3 of the intervention sessions (16 of 24 sessions) were included in the statistical analysis. Accordingly, 32 participants were included in the final analysis (Figure 1). All participants gave their written informed consent to participate in the study, which was in accordance with the Declaration of Helsinki and approved by the local authorized ethics committee. 

### 2.2. Intervention

#### Physical Activity Break

The participants in the intervention group performed the physical activity break, which mainly consisted of juggling exercises, for 15 to 20 min twice a week for 12 weeks (24 sessions). After the physical activity break was introduced to the participants by a sport scientist, they performed the physical activity breaks on their own via online training videos. They always started with a short warm-up, which included mobilization exercises (~1 min). Then, they practiced the juggling exercises (~10–15 min.). At the beginning of the intervention period, the participants started the juggling training with easy throwing and catching exercises with a scarf, bean bag, or ball. Progressively, the difficulty of the juggling exercises was increased by introducing more difficult tasks and using more scarfs, bean bags, and balls at a time, to the maximum of three. The participants were informed that the exercises may be repeated in the next session(s), depending on the difficulty of the exercise. This was done to remove the pressure to learn one exercise within one session and to practice outside of the physical activity breaks. The session ended with a relaxation exercise (~2–3 min). Once a week, the participants performed a static or dynamic balance task (~3 min) before the relaxation part. The participants had the opportunity to practice the balance exercise on the floor, or, if they chose to make the task more challenging, on a balance pad. To offer a variety of balance exercises, the participants trained a new exercise every week (Table 1). After each practicing session, the participants had to fill out a questionnaire to provide feedback on the performed training.

**Table 1 ijerph-20-02193-t001:** Examples for the physical activity break sessions for the three different weeks of training.

	**1. Session**	**2. Session (of 24)**
	** *Exercises* **	
**Mobilization**	Mobilization of the shoulders Backwards rotation of the shoulders	Mobilization of the spine Flexion and extension of the back
**Juggling**	1 scarf 1. “Throwing and catching” with the same hand 2. “Diagonal throwing and catching” Throw the scarf up to the diagonal side and catch it by the other hand 2 scarfs 3. “Throwing and catching” alternately with both hands 4. “Throwing and catching” at the same time with both hands 5. “Diagonal throwing and catching” 2 bean bags 6. “Throwing, one round around the bean bag, catching” Throw the bean bags. Then each hand makes a circle around the thrown bean bag and catches it again.	1 ball 1. “Throwing and catching” with the same hand 2. “Throwing, clapping, and catching” While the thrown ball is in the air, clap with your hands. 3. “Throwing, touching the thigh, and catching” While the thrown ball is in the air, touch the front of your thighs. 4. “Throwing, clapping behind the back, and catching” While the thrown ball is in the air, clap with your hands behind your back. 5. “Line” (Figure 2) 3 scarfs 6. “Diagonal throwing and catching” 7. “3-scarfs-cascade”
**Balance**	Standing on one leg and rotating the head to the left and right side	-
**Relaxation**	“Mindfulness breathing” Perceive yourself breathing in and out.	“Chest breathing” Lay your hands on your chest and make yourself aware of breathing in and out.
	**9. Session**	**10. Session (of 24)**
	** *Exercises* **	
**Mobilization**	Mobilization of the spine and the pelvis Flexion and extension of the back and the pelvis	Mobilization of the ankle Rotation, flexion, and extension of the feet while sitting on a chair
**Juggling**	3 scarfs 1. “3-scarfs-cascade” 2 balls 2. “Throwing, ½-turn, and catching” 3. “Diagonal throwing and catching” 3 balls 4. “3-ball-cascade”	3 scarfs 1. “3-scarfs-cascade” 2. “3-scarfs-cascade with Flash” During the cascade movement, throw each third scarf much faster (= “Flash”) than during the normal movement. 3. “3-scarfs-cascade with Flash and moving around” Perform the exercise and turn one round (longitudinal axis). 2 balls 4. “Wiper plus” (Figure 2) 1 ball 5. “Challenge” Throw the ball with the right hand over your right shoulder and catch the ball behind your back with the right hand. Then throw the ball over the left shoulder and catch it with the left hand.
**Balance**	Standing on one leg with or without the balance pad	-
**Relaxation**	“Breathing in and out with back-movements” Breathe in while extending your back and breathe out while bending your back.	“Mindfulness breathing” Try to breath out longer than you have breathed in.
	**17. Session**	**18. Session (of 24)**
	** *Exercises* **	
**Mobilization**	Mobilization of the pelvis Flexion and extension the pelvis while sitting on the balance pad on a chair	Mobilization of the shoulders Forwards and backwards swinging of the arms
**Juggling**	1 scarf 1. “Catching the scarf with the foot” Throw the scarf and catch it with your foot before or behind your body. 3 scarfs 2. “Column with both hands” Throw two scarfs at the same time, one scarf in front of your body while the other two scarfs are in the air. Catch the first two scarfs, and then the third scarf. Then try to maintain the movement without stopping. 3. “Backwards-cascade” The movement is the same as with the cascade, but you throw the balls from the outside to the inside. 2 balls 4. “Column with one hand” Alternately throw and catch two balls in one hand.	2 balls 1. “Statue of Liberty” (Figure 2) 3 bean bags 2. “3-bean bags-cascade”. 3. “3-bean bags-cascade with body stop” Stop the cascade movement by catching the last thrown bean bag between your elbow/knee and the hand of the other side. 3 scarfs 4. “Column with both hands”
**Balance**	Lunges forwards, sidewards, and backwards, with or without a balance pad	-
**Relaxation**	“Juggling ball massage” Massage your chest, shoulder, and arm muscles with a juggling ball with a comfortable pressure.	“Sun salutation” – short form

**Figure 2 ijerph-20-02193-f002:**
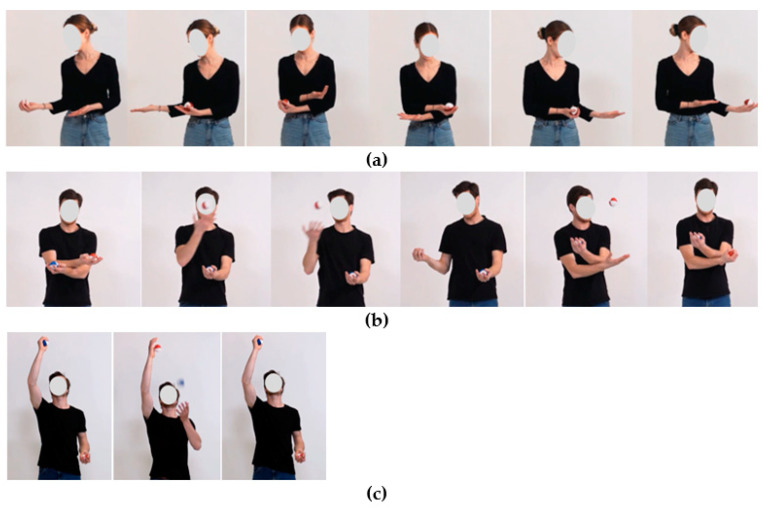
Exercises (**a**) “Line”, (**b**) “Wiper plus”, and (**c**) “Statue of Liberty”.

### 2.3. Assessments

#### 2.3.1. Demographic Data and Physical Activity

At all three measurements sessions, demographic data were collected with a questionnaire. The questionnaire was also used to assess the physical activity of the participants. The participants were asked how many minutes per week they performed moderate physical activities, vigorous physical activities, and strengthening exercises in the last six weeks. The six-week period was chosen because this was the time period in between the measurements. Furthermore, they were asked how much they enjoy physical activity and sports, in general, on a scale from 1 to 4 (1—I do not enjoy physical activity and sports; 4—I enjoy physical activity and sports very much).

#### 2.3.2. Motor Abilities

The Purdue Pegboard Test (Lafayette Instrument Company, Lafayette, IN, USA) was used to assess unimanual and bimanual finger and hand dexterity. During this test, the participant tries to put as many pins as possible in holes of a test board within 30 s. First, the participant performs the exercise with the dominant hand, then with the non-dominant hand, and then with both hands. Each condition was performed three times for 30 s. The averages of the three trials of each condition were used for the statistical analysis. Furthermore, the average of the three scores (dominant, non-dominant, both hands = Σdnb) was calculated as additional score [28].

To measure the reaction time, the Fall Stick Test was used on both hands. The fall stick apparatus was a wooden 1.30 m-long rod with measurement markings. A rubber disc with a diameter of 7.5 cm and a weight of 280 g was connected to the lower end of the stick to stabilize the vertical fall of the stick. Before the start of the test, the participant sat at the side of a table and put his/her forearm with an open hand on the table. The open hand was at the edge of the table. The fall stick apparatus was positioned vertically by an examiner over the participant’s hand, right between the thumb and the index finger without touching the rubber disc. The disc was aligned with the top of the participant’s hand. The examiner released the apparatus in random time intervals ranging from 4 to 10 s, and the participant had to catch the apparatus as quickly as possible. When the participant caught the stick, the distance from the top of the disc to the most superior part of the participant’s hand was recorded. The participant started the test with the dominant hand.

After three practice trials, the participant performed eight experimental trials. For analysis, the best- and worst-performed trial were excluded and the average of the remaining six trials was used for further analysis [29,30].

The dynamic balance performance was assessed with the Y Balance Test (Y Balance Test Kit™). The Y Balance Test consists of a stance platform from which three PVC pipes are attached in the anterior, posteromedial, and lateral directions. A moveable box is attached to each of the pipes. While standing with one leg on the platform, the participant had to push the box as far away from him/herself as possible. The test procedure started with four practice trials [31] for each leg and direction, followed by a five-minute break. During the break of the first measurement session, the leg length of the participant was measured from the spina iliaca anterior to the medial malleolus. This was needed to normalize the reached distance to the leg length. After the break, the participant performed three experimental trials for each leg and direction. The average of the three trials was used for the statistical analysis [32].

#### 2.3.3. Juggling Performance

To assess the juggling skills of the participants, the time (seconds) of juggling a 3-ball-cascade was measured. The participants performed the 3-ball-cascade five times for as long as possible. Before the five test trials, the examiner showed the 3-ball-cascade to the participants. The time from the best performance was used for the analysis.

#### 2.3.4. Evaluation of the Intervention

After each training session, the participants had to answer a short questionnaire (maximum of 6 questions), designed to gain insight into whether the participants were able to accomplish the planned training progression or if there was a need to change the training content. We asked the questions “Was the exercise xxx explained/demonstrated understandable for you?” (yes/no) and “Can you now perform the exercise xxx?” (1- “I agree” to 5- “I disagree”). Furthermore, we asked how they enjoyed the session overall (1- “I enjoyed it very much” to 5- “I did not enjoy the training”), if they were in a positive mood after the training (1- “I agree” to 5- “I disagree”), and when (time) the activity break was performed. At the end of each month, the participants had one session in which all new learned exercises of that month were performed again. After this session, we asked them if they were able to perform those exercises (1- “I agree” to 5- “I disagree”). Overall, 23 juggling exercises were learned by the participants. 

Besides this evaluation of each session, the participants were asked to answer an evaluation of the intervention after finishing all the physical activity breaks. With this evaluation, we examined how they enjoyed the intervention overall (1-“I enjoyed it very much” to 5- “I did not enjoy”), how they enjoyed the different parts of one session (e.g., “Do you agree, that the relaxation exercises were comfortable and relaxing for you?”) and if the active breaks presented were diversified and lively (1- “I agree” to 4- “I disagree”). In addition, we wanted to know if they planned to continue practicing the exercises following the study participation (yes/no). Furthermore, they were asked if they think that the performance of the active breaks had a positive effect on their working routine and health (1- “I agree” to 4- “I disagree”). Finally, we wanted to know if they noticed an increase in their physical activity behaviour (yes/no) and, if yes, how they observed this change in their behaviour.

### 2.4. Statistical Analysis

For statistical analysis, SPSS Statistic 27 (SPSS Statistics; IBM, 2021) was used. All analyses were performed after checking for normal distribution and sphericity of the data and the level of significance was set to 5%. Analyses of variances (ANOVAs) for repeated measurements were used when the data was normally distributed. We used the Greenhouse–Geiser correction if Mauchly’s test was significant. If a significant interaction effect (time × group) was observed, post hoc tests were performed via paired and unpaired *t* tests within and between groups, respectively. Estimates of effect sizes are given in terms of partial eta-squared measures (η2p). G*Power was used to assess the effect size for the *t* tests (Cohen’s d). If the data was not normally distributed, the Friedman, Wilcoxon, and Mann-Whitney U Test were applied to analyse the data. Moreover, for analysing the juggling performance data, we excluded two participants of the control group who were already skilled jugglers. Due to the not normally distributed juggling performance variable, the Spearman’s correlation was executed to analyse the relationship between the changes (over six and twelve weeks) in the motor ability variables and the changes (over six and twelve weeks) in the juggling performance variable of the intervention group. For the evaluation, descriptive data analyses were used.

## 3. Results

### 3.1. Demographic Data and Physical Activity

A total of 32 participants (25 females, 7 males) finished the study. Twenty participants (15 females, 5 males) were in the intervention group, and twelve participants (10 females, 2 males) served as the control group. The distribution of sexes was not significantly different across both experimental groups (χ² = 0.305, *p* = 0.581). There were not any significant differences in the variables age (F(1.4,40.8) = 0.19, *p* = 0.742, η^2^*p* = 0.006), BMI (t [30] = 0.245, *p* = 0.808) and enjoyment of physical activity and sports (χ² = 1.415, *p* = 0.702) (Table 2). Furthermore, no differences between the amount of moderate or high physical activity or strengthening training were found for any of the three measurement timepoints between the groups. There was a within-group difference for the intervention group for the amount of strengthening training (IG: χ²(2) = 9.591, *p* = 0.008). The amount of strengthening training decreased significantly between week 6 and week 12 (54.47 ± 86.01 vs. 36.58 ± 60.69 min/wk; z = −2.075, *p* = 0.038) and week 0 and week 12 (69.05 ± 105.16 vs. 36.58 ± 60.69 min/wk; z = −2.490, *p* = 0.013). There were no significant time differences for the other physical activity parameters for both groups (Table 3).

### 3.2. Motor Abilities

No significant interaction effects (time x group) were found for finger and hand dexterity (dominant hand: F(2,60) = 2.23, *p* = 0.116, η^2^*p* = 0.069; non-dominant hand: F(2,60) = 0.94, *p* = 0.395, η^2^*p* = 0.030; both hands: F(2,60) = 1.24, *p* = 0.297, η^2^*p* = 0.040; Σdnb: F(2,60) = 0.92, *p* = 0.406, η^2^*p* = 0.030), the reaction time (dominant hand: IG: χ²(2) = 1.241, *p* = 0.538; KG: χ²(2) = 0.667, *p* = 0.717; IG vs. KG: week 0: z = −1.324, *p* = 0.186, week 6: z = −0.837, *p* = 0.402, week 12: z = −0.973, *p* = 0.346; non-dominant hand: F(2,60) = 1.74, *p* = 0.185, η^2^*p* = 0.055), or the dynamic balance (RA: F(2,60) = 0.78, *p* = 0.760, η^2^*p* = 0.009; LA: F(2,60) = 0.73, *p* = 0.487, η^2^*p* = 0.024; RPM: F(2,60) = 1.45, *p* = 0.0242, η^2^*p* = 0.046; LPM: F(1.6,49.4) = 0.23, *p* = 0.0960, η^2^*p* = 0.001; RPL: F(1.7,50.3) = 0.83, *p* = 0.425, η^2^*p* = 0.027; LPL: F(2,60) = 1.12, *p* = 0.333, η^2^*p* = 0.036).

### 3.3. Juggling Performance

The analysis of the juggling performance (χ²(2) = 36.747, *p* < 0.001) resulted in a significant increase for the intervention group between week 0 and week 6 (0.20 ± 0.89 vs. 1.63 ± 1.49 s; z = −3.982, *p* < 0.001), week 6 and week 12 (1.63 ± 1.49 vs. 3.82 ± 4.13 s; z = −3.417, *p* < 0.001), and week 0 and week 12 (0.20 ± 0.89 vs. 3.82 ± 4.13 s; z = −3.935, *p* < 0.001). For the control group, no significant change in the juggling performance was detected. In addition, the Mann-Whitney U Test showed a significant difference between the intervention and control groups for week 6 (U = 0.0, *p* < 0.001) and week 12 (U = 0.0, *p* < 0.001) but not for week 0 (U = 95.0 *p* = 0.846) (Figure 3).

### 3.4. Correlations between Changes in Motor Abilities and Juggling Performance

No significant relationships were observed following 6 weeks of training. However, after 12 weeks, a strong negative correlation was found between the changes in reaction time for the dominant hand and the changes in juggling performance (r(18) = −0.617, *p* = 0.004) (Figure 4). Furthermore, changes in finger and hand dexterity and dynamic balance were not related to changes in juggling performance over 6 or 12 weeks.

### 3.5. Evaluation of the Intervention

All participants agreed that 15 out of the 20 exercises were demonstrated understandably. Only four out of the 20 participants mentioned that one or two exercises were not explained understandably. These participants were contacted and got further instructions for these exercises. In summary, 98.4% of the 20 exercises were explained satisfactorily. The analysis of the question “Can you now perform the exercise xxx?” for the 23 juggling exercises showed that 68.4% agreed, 20.9% rather agreed, 7.2% rather disagreed, and 3.5% disagreed. The overall enjoyment of all sessions was rated with a mean value of 1.56 ± 0.16. The analyses of the question “Are you now in a positive mood?” led to a mean value of 1.38 ± 0.21. Most physical activity breaks were performed around noon (28.4%), before noon (22.7%), and in the afternoon (21.5%), while 14.9% of the sessions were performed in the evening, 6.5% at night, and 6.1% in the morning.

The evaluation after finishing all physical activity breaks was completed by sixteen participants (women = 12, men = 4) of the intervention group (Appendix A). The overall enjoyment of the active breaks was rated with 1 (best value) by thirteen participants (81.3%), with 2 by two participants (12.5%), and with 3 by one participant (6.3%).

All participants agreed (93.8.%) or rather agreed (6.3%) that the physical activity breaks presented were diverse and lively. The mobilization and relaxation parts were enjoyed by all participants. When we asked them how challenging the juggling exercises were for them, seven participants (43.8%) found the exercises neither too challenging nor not challenging, five participants (31.3%) indicated that the juggling exercises were too challenging, and four participants (25.0%) were both challenged by some exercises and not challenged by other exercises. The balance tasks were neither too challenging nor not challenging for ten participants (62.5%). For two participants (12.5%), the balance tasks were challenging, not challenging, or too challenging as well as not too challenging, respectively. Most of the participants agreed that the physical activity break had a positive effect on their health (I agree: 11, 68.8%; I rather agree: 4, 25.0%; I rather disagree: 0, 0.0%; I disagree: 1, 6.3%) and their working routine (I agree: 11, 68.8%; I rather agree: 2, 12.5%; I rather disagree: 2, 12.5%; I disagree: 1, 6.3%). Ten (62.5%) participants confirmed that they increased their physical activity. Six participants reported being more physically active before, during, or after work, and two participants had started using the stairs instead of the elevator. The other two participants did not explain how they increased their physical activity. Everyone indicated a motivation to continue with the exercises. Fifteen (93.8%) wanted to continue the juggling, ten (62.5%), the mobilization exercises, six (37.5%), the balance tasks, and four (25.0%), the relaxation exercises.

## 4. Discussion

The regular engagement of the young and middle-aged adults in the three-month physical activity break was associated with positive effects on the juggling performance. Furthermore, we found a relationship between the performance changes in the juggling skills and performance changes in the reaction time. Accordingly, the more the participants increased their juggling performance, the more they improved their reaction time. In addition to the improvements in the juggling performance, the participants enjoyed the physical activity breaks very much and related that their participation had positive effects on their health and working routines.

Although the participants in the intervention group improved their juggling skills, our participants were not very skilled jugglers after completing the intervention (3-ball- cascade time: 3.75 ± 4.23 s) compared to data in previous published juggling studies, where the participants were able to reach times between 20 to 180 s [33,34,35,36]. However, in these studies, participants trained only the 3-ball-cascade. In the present study, we integrated different kinds of juggling exercises, including the 3-ball-cascade, to offer variety during the physical activity breaks. Furthermore, it was the idea that the participants learn different exercises to perceive their improvements and have a feeling of success to keep them motivated to continue to participate in the physical activity breaks. In addition, compared to other studies [33,34,35,36], we did not set a specific goal (e.g., time for the 3-ball-cascade) that the participants should accomplish at the end of the intervention period, but we did have a specified training time. Therefore, we also had a larger variation in juggling performance in our study. Nevertheless, we found that the improvements in the reaction time of the dominant hand correlated significantly with the increases in the juggling performance. To our knowledge, this is the first study investigating the effects of juggling training on motor abilities leading to this novel result.

Despite significant increases in juggling skills, motor abilities like finger and hand dexterity, reaction time, and dynamic balance did not change significantly. Although the intervention group somewhat increased their manual finger and hand dexterity, assessed with the Purdue Pegboard Test, the changes did not reach significance. This could be related to the fact that the participants already accomplished average results at the Purdue Pegboard Test at week 0 [28]. Moreover, studies investigating upper body (resistance) training [37,38] in people with impairments (e.g., multiple sclerosis, essential hand tremor) showed that the healthy side of the hand needs more time to improve its hand dexterity [38]. This would explain the lack of significant changes in the present study with a healthy population. Other researchers did indeed report that a specific training for the hand/fingers (e.g., training at the Pegboard apparatus, sensory awareness training, training with a putty, Wuqinxi exercise) can improve finger and hand dexterity in healthy younger adults [39,40] and older adults with Parkinson disease [41,42]. In the present study, the intervention might have been not specific enough to elicit such changes.

Reaction time, assessed with the Fall Stick Test, decreased for the intervention group, without reaching a statistical significance. A similar result was reported by Ordnung et al. [43], who could not find a statistically significant change in the reaction time, assessed with a Fall Stick Test, after a 6-week exergaming training, despite increased gaming performance in the intervention group. However, in other studies, the reaction time could be improved after long-term interventions (6 to 12 weeks) of different kinds of physical activities in young adults [44] and older women [45]. However, these studies used computerized programs to assess the reaction time, which could be more sensitive to changes than the Fall Stick Test.

The participants of the intervention group did improve their dynamic balance over time in all directions of the Y Balance Test, but this was also observed in the control group. In both groups, the changes were not statistically significant. Despite following the instruction for the Y Balance Test as commonly used [32], and including a familiarization session for the motor ability tests, the results seem to be caused by a learning effect. The duration of the balance training, which was around three minutes per week, seems to have been not enough to induce balance improvement. In other studies, in which improvements were found after balance training at the workplace, the training volume was 60 [19], 90 [46], and 105 [47] minutes per week, which was much higher compared to our study. As shown in a study of Cloak et al. [48], it is possible that after a low-volume balance training (3 min), acute positive effects on dynamic balance can be found. They measured the dynamic balance directly after the balance training session and observed an increase in dynamic balance in the anterior direction. Therefore, it is possible that our balance exercises improved dynamic balance, but it could not be detected since we did not measure dynamic balance directly after the subjects performed the exercise.

The participants agreed that the physical activity breaks were enjoyable and were diversified and lively. This is important for the compliance and continuation of the training. Only participants who accomplished 67% of the intervention (16 out of 24 sessions) were included in the analysis. In similar studies, the required minimal attendance rate was between 67 and 70% [49,50,51]. Although we set the attendance at 67%, the participants reached an average attendance rate of 97.4%, which is a very positive result and supports the result that the participants enjoyed the physical activity breaks. Furthermore, we also saw a low dropout rate of 11.1% in our study. In studies with interventions with coordinative exercises [50,51,52] or physical activity interventions at the workplace [47,49,53,54,55,56], the dropout rate was between 0.0% to 53.8%. It appears that our strategies for maintaining the intervention (e.g., the opportunity to contact us by email or by phone, upload a video with questions about the training progress, personal coaching at the office, talk/discussion about the training progress at the measurements) were successful. In addition, the enjoyment and high number of practiced sessions led to improvements in the participants’ subjective health and working routines. These results are in accordance with other studies, where an increase in the health- and work-related parameters of participants in an workplace setting were found after exercise interventions [5,6,7,8,9,10]. When the participants of this study explained how they observed a positive influence of the intervention on their health, they described an improvement of their physical and mental health (Appendix A). Two participants also mentioned that their neck pain decreased after performing the physical activity breaks. This is in accordance with the results of a study of Sjøgaard et al. [57], where office workers reduced their neck pain by integrating aerobic and strength exercises during their working hours. Furthermore, our participants subjectively observed improvement in their motor abilities, which they related to an improvement in their health. Although these subjectively recognized improvement of the motor abilities cannot be statistically confirmed by the results of the motor abilities tests, it seems that the participants perceived an improvement in their motor abilities due to performance progression of their juggling skills and balance during participating in the physical activity breaks. These are important results in relation to sickness absenteeism, since the second-most frequent cause of sickness absenteeism in Austria is musculoskeletal disorders. Additionally, an improvement in mental health is important due to the fact that mental disorders are causing the longest periods of sickness absenteeism [58]. Moreover, approximately sixty percent of Austrian working adults indicated having psychological stress at work [59]. Therefore, it is especially important to address mental health at the workplace. In this study, the participants observed a positive effect of the physical activity breaks on their working routine. They mentioned that the physical activity breaks led to increased concentration, added variety to their working routine, and helped them to enhance their awareness of the importance of taking breaks during the working hours (Appendix A). Although these are subjectively observed changes in a small sample, it gives at least an insight into how physical activity breaks can influence employees’ health and working routines. Furthermore, it is known that people can become more physically active the more they trust its positive effect on their health. A perceived self-efficacy regarding the positive influence of physical activity could mediate healthy behaviour [60]. In this study, more than 60% of the participants described that they subjectively increased their physical activity, although these results cannot be supported by the results of the physical activity questionnaire. In summary, it can be assumed that a physical activity break can help to prevent health- and work-related declines and to improve physical activity behaviour. From a public health perspective, these are important findings, since it is possible to reach a large group of the population (15 to 65 years) by implementing physical activity breaks at the workplace. Additionally, due to the backwards shifting of the point of retirement entry, the working population is getting older [61]. Therefore, maintaining ordinary daily physical functions and the ability to move by participating in physical activities and sport is important for those aging workers. Moreover, healthy adults are playing a major rule in the public health context [57].

Furthermore, we can conclude that the organization and implementation of the physical activity breaks in an online format were sophisticated and successful. Firstly, a major benefit of the physical activity break was the easy feasibility of the intervention. The participants could practice at any time during their working hours at their office and didn’t need to change their clothes to do the training or shower afterwards. Secondly, due to some restrictions, which did not allow working fulltime in the office (COVID-19) during the intervention period, the participants had to work from home. Because we were using an online platform and a small number of necessary materials for the intervention, it was still possible for the participants to perform the physical activity breaks at home. This might be important for future interventions, as the proportion of home office hours has increased in the last years. Moreover, the evaluation questionnaire after each physical activity break allowed us to monitor whether the sessions were performed regularly by the participants and if there was a need to adapt the planned progression of the exercises.

Nevertheless, this pilot study also has some limitations. Regarding the results of the physical activity assessment, nearly all our participants achieved the Austrian physical activity recommendations. Therefore, our participants were already very active at the start of the intervention. This could be a reason why we have not found significant changes in the motor ability variables. Another reason could be the training volume of the physical activity breaks. For this physical active sample, it is possible that the training volume (~26.5 min of coordination training/week) was too low to lead to improvements in motor abilities. Although no participant of the intervention group was regularly practicing intense coordinative and/or motor training, they mentioned different kinds of physical activities (e.g., dancing, martial arts, yoga, cross-country skiing, climbing) in the physical activity questionnaire that require a specific amount of coordinative ability to be performed. Therefore, it is possible that the sample of this study already had a higher level of motor fitness. In addition, our sample had a positive approach to physical activity and was enjoying doing sports. This might have been a reason to volunteer for the present study, as it was known to be a challenge to address inactive employees and employees with a higher risk for poor health [62,63]. A further limitation is the imbalanced number of women and men who participated in the study. This is, however, in line with other studies, where the percentage of women was between 64.2% and 90.0% [49,54,55,56,64,65]. Women are more likely to participate in worksite programs [66], and men and women do have different motives to be physically active [67]. Although we found out that the intervention had a positive impact on the participants’ working routines, we did not assess the influence on their productivity. For example, Santos et al. [18] evaluated the productivity of dairy industry workers with the Health and Work Performance Questionnaire, and they found an increase in the participants’ productivity. However, such an analysis was not possible in the present study due to great heterogeneity in the working tasks of the participants. There is another limitation regarding the balance part of the physical activity break; we did not control the use of the balance pad during the balance exercises. We instructed the participants to use a balance pad to increase the difficulty of the exercise without making it too challenging. Although we could not assess the difficulty of the exercises objectively, the results of a study of Blasco et al. [68] had shown that, independent of standing on stable or unstable ground, balance training can improve balance performance.

## 5. Conclusions

To conclude, the 12-week physical activity break with coordinative exercises led to improvements in the juggling performance and subjective health and working routines, without any changes in general motor abilities like reaction time, finger and hand dexterity, or dynamic balance. In addition, the web- and video-based physical activity breaks had good feasibility and were rated as enjoyable by the participants. Therefore, this pilot study shows that it is possible to implement physical activity breaks at the workplace. The implementation helps to reach a large group of the population (16 to 65 years). Furthermore, it offers an opportunity to be physically active within the working hours. These aspects are important from a public health perspective. For further research, we recommend the implementation of physical activity breaks with a sedentary sample and/or a higher training volume (e.g., 15 to 20 min 3 to 5 times per week).

## Figures and Tables

**Figure 1 ijerph-20-02193-f001:**
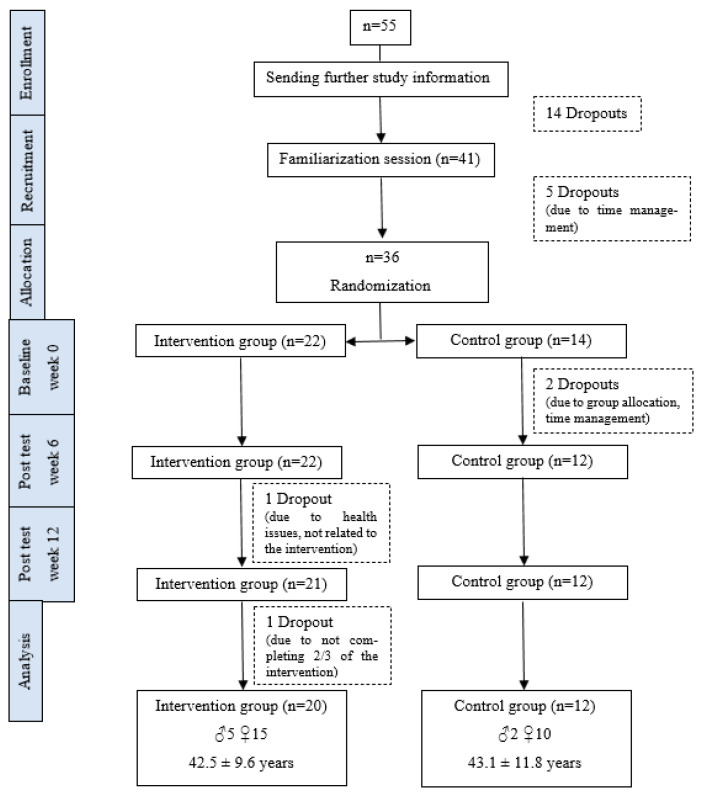
Flow diagram of participants.

**Figure 3 ijerph-20-02193-f003:**
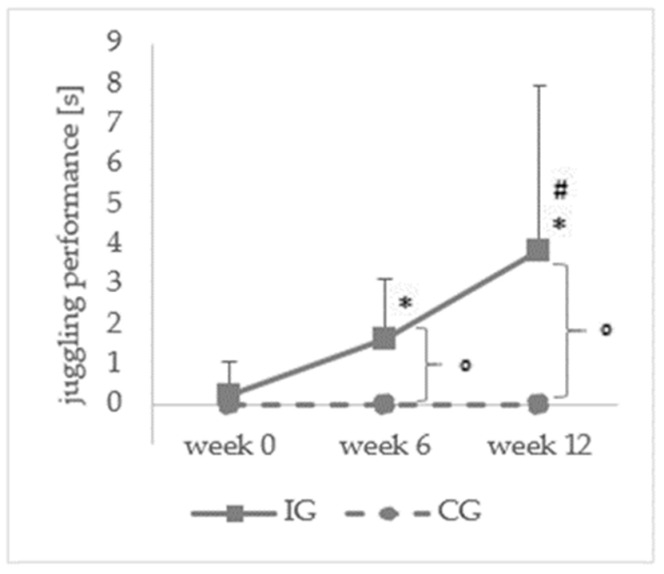
Means and standard deviations of the 3-ball-cascade absolute time before (week 0), during (week 6), and after (week 12) for the intervention group. * significant difference to week 0 for the IG, # significant difference to week 6 for the IG; ° significant difference between IG and CG; KG: n = 10.

**Figure 4 ijerph-20-02193-f004:**
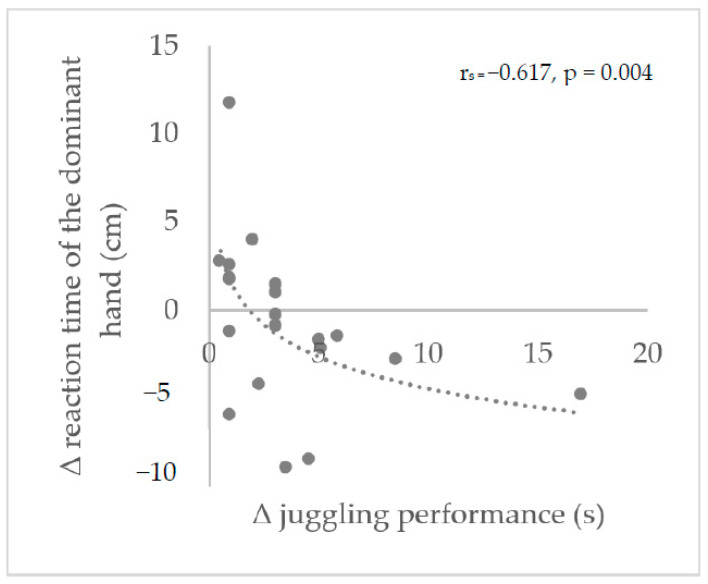
Spearman’s correlation of the changes over 12 weeks of the 3-ball-cascade absolute time and the changes in reaction time over 12 weeks of the absolute distance of the dominant hand at the Fall Stick Test. The negative relationship between Δ the reaction time and Δ the juggling performance represent a positive result: the greater the reduction of the reaction time, the greater the increase in juggling performance.

**Table 2 ijerph-20-02193-t002:** Demographic data of the participants of the intervention and control groups at week 0.

	Intervention Group	Control Group	*p*-Value
Gender (female), n (%)	15 (75.0%)	10 (83.3%)	*p*_Chi_ > 0.05
Age (years)	42.50 ± 9.57	43.17 ± 11.75	Pt > 0.05
BMI (m²/kg)	22.96 ± 2.55	22.74 ± 2.26	Pt > 0.05
Enjoyment of physical activity and sports	3.60 ± 0.82	3.55 ± 0.69	*p*_Chi_ > 0.05
*p*_Chi_ = *p*-Value of χ²; *p*t = *p*-Value of *t* test.			

**Table 3 ijerph-20-02193-t003:** Juggling performance, motor abilities, and physical activity of the participants of the intervention and control groups.

	Intervention Group (n = 20)			Control Group (n = 12)		
	week 0	week 6	week 12	week 0	week 6	week 12
	**Juggling performance (seconds)**					
3-ball-cascade	0.20 ± 0.89	1.63 ± 1.49 *°	3.82 ± 4.13 *^#^	0.00 ± 0.00 °	0.00 ± 0.00 °	0.00 ± 0.00 °
	**Purdue Pegboard Test (number of pegs)**					
dominant	14.77 ± 1.31	15.04 ± 1.33	16.00 ± 1.82	15.55 ± 1.25	15.80 ± 1.37	16.05 ± 1.10
non-dominant	14.22 ± 1.18	14.40 ± 1.32	14.58 ± 1.47	14.31 ± 1.26	14.80 ± 1.70	15.19 ± 1.68
both	11.78 ± 0.98	12.51 ± 1.33	12.61 ± 1.23	12.36 ± 0.90	12.72 ± 1.09	12.61 ± 1.38
Σdnb	40.77 ± 2.81	41.98 ± 3.78	43.18 ± 3.92	42.25 ± 2.80	43.27 ± 3.47	43.55 ± 4.08
	**Fall Stick Test (cm)**					
dominant	27.53 ± 4.70	27.66 ± 4.02	26.75 ± 4.52	25.37 ± 4.73	26.73 ± 4.11	25.79 ± 4.08
non-dominant	27.06 ± 3.93	27.69 ± 4.33	25.43 ± 3.98	25.50 ± 4.23	28.06 ± 4.24	26.21 ± 3.55
	**Y Balance Test (%)**					
RA	59.67 ± 4.96	60.92 ± 5.13	60.97 ± 5.78	58.98 ± 6.03	60.94 ± 7.14	60.47 ± 7.06
LA	59.93 ± 5.05	60.50 ± 4.67	61.69 ± 4.79	59.97 ± 7.34	61.54 ± 6.63	62.57 ± 6.70
RPM	94.95 ± 10.21	97.40 ± 8.63	99.58 ± 7.78	96.87 ± 8.40	97.50 ± 8.00	98.55 ± 8.55
LPM	97.66 ± 8.58	98.98 ± 8.70	100.89 ± 8.21	96.85 ± 10.50	98.55 ± 8.47	100.31 ± 8.54
RPL	92.79 ± 9.21	96.17 ± 8.63	98.70 ± 7.84	94.38 ± 8.05	96.68 ± 8.10	98.30 ± 7.43
LPL	93.71 ± 8.75	96.58 ± 8.06	98.31 ± 7.17	94.65 ± 9.17	96.32 ± 8.07	97.09 ± 7.25
	**Physical Activity (minutes/week)**					
Moderate intensity	298.00 ± 185.57	318.50 ± 256.99	275.00 ± 232.12	431.36 ± 478.94	408.33 ± 406.01	420.00 ± 666.65
High intensity	233.50 ± 229.66	209.55 ± 217.74	254.00 ± 272.11	316.36 ± 320.79	152.17 ± 160.65	117.50 ± 97.25
Strength training	65.60 ± 103.51	51.75 ± 84.59	34.75 ± 59.63 *^#^	82.27 ± 99.58	82.92 ± 120.99	48.50 ± 74.09

RA = right anterior direction; LA = left anterior direction; RPM = right posteromedial direction; LPM = left posteromedial direction; RPL = right posterolateral direction; LPL = right posterolateral direction; wk = week; * significant difference to week 0, ^#^ significant difference to week 6, ° significant difference between IG and CG; KG week 0 physical activity n = 11: one participant didn’t answer the questionnaire at week 0).

## Data Availability

The data presented in this study are available on request from the corresponding author. The data are not publicly available for privacy reasons.

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
