# Peer review of "Twelve Weeks of Web-Based Low to Moderate Physical Activity Breaks with Coordinative Exercises at the Workplace Increase Motor Skills but Not Motor Abilities in Office Workers—A Randomised Controlled Pilot Study"

_ijerph, 2023, doi:10.3390/ijerph20032193_

Round 1

Reviewer 1 Report

The authors aimed to evaluate the effects of  physical activity breaks (15-20 min)  at home consisting of  coordinative exercises (PAB) and balance tasks 2-times/week for 12 weeks  on motor abilities in middle-aged healthy employees of Graz University.

Although the argument was interesting, however the limited number of participants, almost all females, and the enrollment of subjects that usually participate in sport and physical activity programs, have limited the results obtained.  

Major revision

In my opinion the authors must evaluate by appropriate test if the improvement in reaction time, observed in the Intervention Group may had a positive impact on the work carried out by the participants. 

In fact, the authors only mention improvements in health and working activities  (lines 406-409) in the discussion; an in deept discussion of this aspect  can improve and increase the interest of the research and results.

Author Response

Dear reviewer,

first of all, we want to thank the reviewer for the constructive criticism. The comments showed us the shortcomings of our manuscript and helped us to understand at which parts readers could have problems. We tried to address each comment on a point-by-point basis in italic format and included changes in the revised manuscript in red font accordingly. Please see our responses below:

Reviewer comment:

Although the argument was interesting, however the limited number of participants, almost all females, and the enrollment of subjects that usually participate in sport and physical activity programs, have limited the results obtained. 

We are aware of the limitations due the limited number of participants, the homogeneity of the sexes, and that the participants were already a physically active sample. Therefore, this is a pilot study, as mentioned in the title, the introduction (line 84) the discussion (line 460), and conclusion (line 498). Furthermore, these limitations are discussed in the discussion from line 460 to 479.

Reviewer comment:

In my opinion the authors must evaluate by appropriate test if the improvement in reaction time, observed in the Intervention Group may had a positive impact on the work carried out by the participants.

We agree that an evaluation of the influence of the improvements on the work carried out would be interesting. But, due to the heterogeneity of the work tasks of the participants, it is difficult to evaluate the effect of the intervention on the work productivity.  Since the study is already finished, it is furthermore not possible to evaluate the influence. However, we evaluated if the participants observed a subjective positive effect of the intervention on their working routine. These results are now included within a supplemental table 4 and mentioned from line 318 to 320 and discussed from line 405 to 408; 426 to 432. Furthermore, we addressed that we did not evaluate the impact on the productivity of the participants as limitation in the discussion from line 479 to 485.

Reviewer comment:

In fact, the authors only mention improvements in health and working activities (lines 406-409) in the discussion; an in-depth discussion of this aspect can improve and increase the interest of the research and results.

We added some further discussion about the improvements in health and working activities from line 408 to 446.

Reviewer 2 Report

A well written and presented paper, rigorous properly designed study. The limitation in appeal is the narrowness in the findings relating to one exercise ´juggling´ and a lack of expansion to its application to a public health context. 

Author Response

Dear reviewer,

first of all, we want to thank the reviewer for the constructive criticism. The comments showed us the shortcomings of our manuscript and helped us to understand at which parts readers could have problems. We tried to address the comment in italic format and included changes in the revised manuscript in red font accordingly. Please see our response below:

Reviewer comment:

A well written and presented paper, rigorous properly designed study. The limitation in appeal is the narrowness in the findings relating to one exercise ´juggling´ and a lack of expansion to its application to a public health context.

We added some further discussion regarding the improvements in health and working activities from line 408 to 446, which also includes the public health context from line 438 to 446. Furthermore, we added the public health perspective in the conclusion from line 497 to 501.

Round 2

Reviewer 1 Report

the authors have responded comprehensively to the criticisms. tHe paper in the revised form (2) is now suitable